# Efficacy and Safety of Rechallenge with BRAF/MEK Inhibitors in Advanced Melanoma Patients: A Systematic Review and Meta-Analysis

**DOI:** 10.3390/cancers15153754

**Published:** 2023-07-25

**Authors:** Jonathan N. Priantti, Maysa Vilbert, Thiago Madeira, Francisco Cezar A. Moraes, Erica C. Koch Hein, Anwaar Saeed, Ludimila Cavalcante

**Affiliations:** 1School of Medicine, Federal University of Amazonas—UFAM, Manaus 69020-160, AM, Brazil; 2Princess Margaret Cancer Centre, University Health Network, Toronto, ON M5G 2M9, Canada; 3Division of Medical Oncology, Department of Medicine, University of Toronto, Toronto, ON M5S 1A8, Canada; 4School of Medicine, Federal University of Minas Gerais—UFMG, Belo Horizonte 30130-100, MG, Brazil; 5School of Medicine, Federal University of Pará—UFPA, Belém 66075-110, PA, Brazil; 6Department of Hematology and Oncology, School of Medicine, Pontificia Universidad Católica de Chile, Santiago 8331150, Chile; 7Department of Medicine, Division of Hematology and Oncology, UPMC Hillman Cancer Center, University of Pittsburgh, Pittsburgh, PA 15213, USA; 8Department of Medical Oncology, Novant Health Cancer Institute, Charlotte, NC 28204, USA

**Keywords:** advanced melanoma, targeted therapy, rechallenge, BRAF/MEK inhibitors, MAPK inhibitors

## Abstract

**Simple Summary:**

Approximately 50% of patients with melanoma harbor a BRAF mutation and are eligible for targeted therapy with BRAF/MEK inhibitors (BRAFi/MEKi). Despite a response rate of nearly 70%, more than half of the patients will experience disease progression within a year due to tumor resistance. Rechallenging patients with BRAFi/MEKi has emerged as an alternative for improving response and survival outcomes. This systematic review and meta-analysis aims to investigate the efficacy and safety of this strategy in patients with advanced melanoma.

**Abstract:**

This systematic review and meta-analysis aims to evaluate the efficacy and safety of rechallenging advanced melanoma patients with BRAFi/MEKi. Seven studies, accounting for 400 patients, were included. Most patients received immunotherapy before the rechallenge, and 79% underwent rechallenge with the combination of BRAFi/MEKi. We found a median progression-free survival of 5 months and overall survival of 9.8 months. The one-year survival rate was 42.63%. Regarding response, ORR was 34% and DCR 65%. There were no new or unexpected safety concerns. Rechallenge with BRAFi/MEKi can improve outcomes in advanced melanoma patients with refractory disease. These findings have significant implications for clinical practice, particularly in the setting of progressive disease in later lines and limited treatment options.

## 1. Introduction

Melanoma represents a significant health problem and economic burden [1,2]. According to recent data from the National Center for Health Statistics, 97,610 new cases of melanoma will be diagnosed in the US in 2023, with approximately 8000 deaths [3]. The incidence has been overall stable, with the rate decreasing in men and increasing in older women by 1% per year [3,4]. The highest incidence is seen in Australia/New Zealand, followed by Western Europe, North America, and Northern Europe [2]. Worldwide, the mortality rate is disproportionally higher in transitioning countries, likely due to difficulties in accessing the newest medications [2,5]. The insufficient evidence for screening as a secondary prevention for the early detection of melanoma patients contributes to the disease burden [6,7]. In addition, some patients experience disease progression or are diagnosed with advanced disease, leading to poorer outcomes [8,9].

Approximately a decade ago, the prognosis of metastatic melanoma patients was dismal, and half would die within a year of diagnosis [10]. Available treatments at that time were ineffective, with low and short-lived response with chemotherapy or significant toxicity from agents, such as high-dose interleukin 2 (HD IL-2) [11,12,13,14,15]. Fortunately, the discovery of driver mutations and immune evasion mechanisms led to the development of targeted therapies and immune checkpoint inhibitors [16], and their approval by the Food and Drug Administration (FDA) and other regulatory agencies was considerable in this regard [11,17]. Several targeted therapies were developed, showing superiority compared to the chemotherapy dacarbazine, which was considered a reasonable approach before the FDA approval of Vemurafenib (BRAF inhibitor) and Ipilimumab (anti-CTLA-4) in 2011 [18,19]. The advances in treatments led to a large increase of 258% in the number of individuals living with metastatic melanoma in the last two decades [20].

The BRAF gene encodes different RAF protein isoforms that activate serine/threonine protein kinases in the mitogen-activated protein kinase (MAPK) signaling pathway [21,22]. BRAF mutations are oncogenic drivers that stimulate cell proliferation and growth through this pathway, augmenting the capacity of tumors to progress and spread [23,24,25]. Roughly 50% of melanomas harbor a BRAF mutation, especially the variant V600E, suitable for targeted therapy with BRAF inhibitors [22,26].

Vemurafenib was the first drug used in this context, and since its discovery, other BRAF inhibitors, such as Dabrafenib and Encorafenib, have become available [27,28,29]. Concurrently, the MAP kinase (MAPK) pathway was meticulously explored, and an important resistance mechanism to BRAFi, the downstream activation of MEK, was identified [11,24]. MEK inhibitors (MEKi) were then incorporated to the treatment landscape. Moreover, clinical studies have demonstrated that the combination of BRAF and MEK inhibition enhances the MAPK pathway blockade, resulting in better and consistent outcomes than BRAFi alone [30,31,32,33,34,35,36,37,38,39]. Surprisingly, this combination also exhibits a better safety profile than either BRAFi or MEKi monotherapy [39,40,41], also showing improvement in health-related quality of life in unresectable III/IV stage melanoma patients [42]. However, despite the advantages offered by targeted therapy with BRAFi/MEKi combination, more than 50% of patients still progress within a year of treatment due to the reactivation of the MAPK pathway through alternative pathways [11,18,43].

Immunotherapy with immune checkpoint inhibitors has been the treatment of choice when targeted therapy fails due to its long-term survival benefit, especially in complete responders [18,44,45,46]. Unfortunately, almost half of the patients will not respond due to primary resistance mechanisms, and most responders may ultimately progress [18,47]. Further treatment options for these patients include a clinical trial, when available, or chemotherapy [48]. More recently, alternatives, such as treatment beyond progression and rechallenge, have stood out [49,50,51,52]. The latter has shown promising results in melanoma, consisting of re-exposure to BRAFi/MEKi after a period on another treatment or a drug holiday [26]. The BRAFi/MEKi-free interval, in theory, allows for sensitive tumor cells to grow, and a subsequent rechallenge could hypothetically induce responses again and improve survival outcomes [53,54]. Given the emerging clinical evidence for this practice, our systematic review and meta-analysis aims to investigate the efficacy and safety of rechallenging advanced BRAF-mutant melanoma patients with MAPK inhibitors.

## 2. Materials and Methods

### 2.1. Eligibility Criteria

Inclusion in this meta-analysis was restricted to studies that met all the following eligibility criteria: (1) randomized and non-randomized clinical trials and prospective and retrospective observational cohort studies; (2) including advanced melanoma patients who previously progressed on BRAFi ± MEKi in 1st or 2nd line therapy; and (3) patients rechallenged with BRAFi/MEKi after other systemic treatments or a drug holiday. In addition, studies were only included if they reported any of the clinical outcomes of interest: survival outcomes, response to BRAFi/MEKi rechallenge, and safety. We excluded studies with the following: (1) no outcomes of interest; (2) case reports, series of cases, or case-control studies; (3) no BRAFi/MEKi rechallenge; or (4) studies with overlapping populations.

### 2.2. Search Strategy and Data Extraction

We systematically searched PubMed, Scopus, and Cochrane Central Register of Controlled Trials databases, and the congresses websites of the American Society of Clinical Oncology (ASCO) publications, European Society for Medical Oncology (ESMO), and American Association for Cancer Research (AACR) from inception to November 2022. The following search terms were used: “melanoma”, “rechallenge”, “disease progression”, “progressed disease”, “re-challenge”, “BRAF/MEK”, “BRAF”, “MEK”, “trametinib”, “cobimetinib”, “binimetinib”, “vemurafenib”, “dabrafenib”, and “encorafenib”. The references from all included studies and previous systematic reviews were also searched manually for any additional studies. Three authors (J.N.P., T.M., and F.C.A.M.) independently extracted the data following the predefined search criteria and quality assessment. If there was no unanimity about including any study, a fourth author decided its eligibility (M.V.).

### 2.3. Endpoints and Subanalysis

Efficacy outcomes included objective response rate (ORR), disease control rate (DCR), progression-free survival (PFS), overall survival (OS), and one-year overall survival rate (1-year OS). ORR included patients with partial or complete response, whereas DCR included patients with stable disease, partial or complete response. Response was evaluated by investigator–clinical assessment or by RECIST (Response Evaluation Criteria in Solid Tumors) criteria version 1.1, according to each study’s own criteria. In the safety analysis, adverse events were characterized based on the Common Terminology Criteria for Adverse Events (CTCAEs) version 4.0. The prespecified analyses of prognostic factors included subgroups of patients with elevated lactate dehydrogenase (LDH) and metastasis to the central nervous system. We sought all results compatible with our outcomes of interest in each study. Then, we built a table containing the study characteristics, interventions, and results and compared them against our planned outcomes to define the groups for each synthesis.

### 2.4. Quality Assessment

Non-randomized interventional studies were assessed through the Risk Of Bias In Non-randomized Studies of Interventions (ROBINS-I) tool [55], which contains seven domains and categorizes studies as having low, moderate, serious, critical, or unclear risk of bias. The quality assessment of observational studies was performed using the Newcastle–Ottawa Scale (NOS), in which studies are scored on a 0 to 9 scale according to selection, comparability, and exposure criteria [56].

### 2.5. Statistical Analysis

The systematic review and meta-analysis were performed in accordance with the Cochrane Collaboration and the Preferred Reporting Items for Systematic Reviews and Meta-Analysis (PRISMA) statement guidelines (PRISMA Checklist, Appendix A) [57]. We conducted a proportional meta-analysis pooling the data with the function “metaprop” and “pool.median”, included in the packages “meta”, “metafor”, and “metamedian” in R for efficacy outcomes [58]. We did not report OS and PFS in hazard ratios (HRs) due to the absence of this information in most studies. Thus, we pooled the medians of these variables through the method proposed by McGrath et al. [59]. The HRs with 95% confidence interval (CI) from multivariate analyses were used to assess prognostic factors, such as elevated LDH and the presence of brain metastasis, using the function “metagen”, also included in the “meta” package in R. Cochran’s Q test and I^2^ statistics were used to evaluate heterogeneity; *p*-values inferior to 0.10 and I^2^ > 25% were considered significant for heterogeneity. A DerSimonian and Laird random-effects model was applied to all analyses. We used RStudio (Posit Software, PBC, version 2022.12.0 + 353) for statistical analysis. The study protocol was registered in the International Prospective Register of Systematic Reviews database—PROSPERO (registration number: CRD42022375952).

## 3. Results

### 3.1. Characteristics of the Included Studies and Patients

The initial literature search generated 1858 results, as detailed in Figure 1. After removing duplicate records and ineligible studies, 34 remained and were thoroughly reviewed based on inclusion criteria. Of these, a total of seven studies were included: one phase II clinical trial and six observational cohort studies comprising 400 patients [60,61,62,63,64,65,66]. A full list of the excluded studies can be found in the Appendix A.

The population was characterized by BRAF-mutant advanced melanoma patients, who had targeted therapy in the first or second line of treatment, with either the combination of BRAFi/MEKi or BRAFi alone. Most patients (83%, 333/400) underwent an interval treatment with immunotherapy before being rechallenged with targeted therapy, and 10% (40/400) went on a drug holiday. During the rechallenge, 79% patients (317/400) were treated with the combination of BRAFi/MEKi, whereas 21% (83/400) received BRAFi alone. The baseline characteristics of the included studies are shown in Table 1, and the summary of the treatments administered during the first targeted therapy, interval treatment, and rechallenge are presented in Figure 2. The description of the best response during the first targeted therapy (TT) exposure (in the first/second line of treatment) and the best response with TT rechallenge in each study are presented in the Appendix A.

### 3.2. Efficacy and Safety of Rechallenging Advanced Melanoma Patients with Targeted Therapy

In a pooled analysis including all seven studies, the ORR was 34.25% (95% CI 28.5 to 40.0) with no significant heterogeneity seen among the studies (I^2^ = 25%, *p* = 0.24), as shown in Figure 3A. The prevalence of DCR, presented in Figure 3B, was 65.01% (95% CI 57.31 to 72.72) with a high level of inter-study heterogeneity (I^2^ = 59%, *p* = 0.02).

Exploring the prevalence of DCR further in a sensitivity analysis, we found that studies with a median BRAFi/MEKi-free interval of six months or more achieved a significantly higher DCR rate than those with a shorter BRAFi/MEKi-free interval (<6 months). The DCR rate was 68% versus 56%, respectively (*p* < 0.01), with a low heterogeneity in each subgroup (I^2^ of 0% and 12%, respectively), as illustrated in Figure 4.

The median progression-free survival (mPFS) during rechallenge with BRAFi/MEKi was 5 months (95% CI 4 to 5.9), based on a pool of medians from five studies. The median OS was 9.8 months (95% CI 9.3 to 20.4), according to data from four studies. The proportion of patients alive in 1 year was 42.63% (95% CI 30.25 to 55.02), with considerable variability among studies (I^2^ = 70%, *p* = 0.01), as depicted in Figure 3C.

We conducted pooled analyses of hazard ratios from three studies assessing PFS and two studies evaluating OS, comparing (1) the presence or absence of brain metastasis and (2) normal versus elevated LDH levels. However, only some studies provided univariate and multivariate analyses evaluating prognostic factors associated with survival outcomes. Consequently, the findings of our combined sub-analyses were also limited. The analyses did not yield statistically significant results. The forest plots can be found in the Appendix A.

Regarding safety, there were no new or unexpected adverse events. Two studies reported that the safety profile during TT rechallenge was very similar to the first TT treatment [60,63], while in one study [66], patients reported a better tolerance during rechallenge. There were no deaths related to treatment. The most frequent toxicities were fever and rash.

### 3.3. Quality Assessment

Our results consist of information collected from one clinical trial and six observational studies. The non-randomized clinical trial was considered to have a moderate risk of bias. All other studies scored 7 out of 9 in NOS, except for one observational study scoring 6 out of 9 due to lack of information on confounding factors [64]. Individual appraisal of non-randomized interventional studies and observational studies can be seen, respectively, in Appendix A.

In the visual inspection of the two funnel plots from the outcomes of DCR and 1-year OS rate, we found a predominantly symmetrical pattern of the number of studies on each side and their disposition around the central axis, as shown in Appendix A. However, two studies were located slightly outside the plot in each analysis, which may represent differences in patient selection and intervention. Aiming to explore it further, we ran Egger’s regression test, which showed no evidence suggesting significant publication bias in either of the two outcomes (Appendix A; DCR: z = 1.7615, *p* = 0.0782; and 1-year OS rate: z = −0. 4770, *p* = 0.6355).

We performed leave-one-out sensitivity analyses by systematically removing each study from the pooled estimates in two variables with high heterogeneity: the prevalence of DCR and the prevalence of 1-year OS rate (Appendix A, respectively). Applying the leave-one-out test did not substantially alter the results. Additionally, when analyzing the influence of some studies on DCR, Roux et al. [34], a small-sized study of nine patients, had the most significant benefit in DCR, although it was the study with the worst 1-year OS rate. When excluded from some of our analyses, we found this study contributed significantly to the high heterogeneity (I^2^ = 0 when removed).

## 4. Discussion

In this systematic review and meta-analysis of seven studies, including 400 patients, we assessed the efficacy and safety of rechallenging BRAF-mutant advanced melanoma patients with BRAFi/MEKi. The main findings related to this therapeutic strategy include the following: (1) BRAFi/MEKi rechallenge is a promising palliative treatment, with more than two-thirds of patients achieving disease control mainly due to partial response and stable disease; (2) an interval therapy longer than 6 months was significantly associated with a greater DCR; (3) objective response was seen in more than one-third of patients; (4) median PFS was 5 months; (5) median OS was 9.8 months, and 43% of patients were alive at 1 year; (6) the presence of brain metastasis or elevated LDH did not show a statistically significant association with an increased risk of progression or death; and (7) patients had a good tolerance, without unexpected adverse events during TT rechallenge.

Since the BRIM-3 trial with Vemurafenib [27,67] and the BREAK-3 trial with Dabrafenib [28,68], BRAF inhibitors have shown greater benefit in improving outcomes for BRAF-mutant unresectable or metastatic melanoma patients compared to chemotherapy with dacarbazine. Moreover, the combination of BRAF and MEK inhibitors has yielded even more promising results, as demonstrated in the COMBI-D phase III trial, combining Dabrafenib with Trametinib [32,33,34,69]. This combination enhanced outcomes by improving the efficacy and reducing the toxicity compared to BRAF monotherapy [70]. Additional clinical trials have also substantiated this approach’s efficacy, as exemplified by the coBRIM trial (Vemurafenib plus Cobimetinib) and the 5-year update of the COLUMBUS trial (Encorafenib plus Binimetinib), which showed consistent efficacy and safety of the BRAFi/MEKi combination [30,31,35,36,37,71]. Unfortunately, most patients will eventually progress despite the encouraging initial responses to treatment [34,43]. This acquired resistance leads to a gradual loss of clinical response during treatment, primarily due to the reactivation of the MAPK/ERK downstream pathway or at the level of the BRAF mutation itself [70,72,73,74,75].

Moreover, recent studies have shown other potential mechanisms involved in response and resistance, such as tumor cell differentiation, epigenetic reprogramming, the production of bioregulatory factors as self-protection against host responses and therapies, and the neuro-endocrine and immune system modulation by melanin pigmentation and melanogenesis to promote tumor growth [76,77,78,79]. In the genetic realm, the upregulation of mutated genes, such as MITF and the receptor tyrosine kinase AXL, as well as downregulation of the tumor suppressor gene PTEN, are associated with BRAF inhibitor resistance [80,81,82]. Additionally, phenotype switching, which confers heterogeneity in MAPK pathway dependence, contributes to sensitivity or resistance to TT and is an area of opportunity for future drug development [83,84,85].

In all studies included in our research, TT was the first-line treatment for metastatic melanoma. Although BRAF inhibition can enhance antigen expression, facilitates T-cell cytotoxicity, and creates a favorable tumor microenvironment, potentially augmenting the efficacy of immunotherapy [86,87,88], recent investigations revealed that resistance to BRAF/MEK inhibitors induces an immune-evasive tumor microenvironment [89,90]. This cross-resistance to immunotherapy in melanoma cells, was characterized by the appearance of non-functional CD103+ dendritic cells in the tumor microenvironment, as a result of MAPK pathway reactivation [90]. As a result, impaired antigen presentation hinders effective T-cell responses and potentially decreases the efficacy of sequential immunotherapy [90]. In contrast, immunotherapy may potentially enhance the response to targeted therapy in BRAF-mutated melanoma, thereby prolonging tumor regression durability [91].

The most current clinical data support the use of ICI as the preferred first-line treatment for BRAF-mutated melanoma [92]. Studies, such as SECOMBIT and DREAMseq, which aimed to determine the optimal initial treatment for these patients, demonstrated that immunotherapy following progression on first-line BRAF/MEK inhibitors yields lower response rates and worse survival compared to the reverse treatment sequence [93,94,95]. All these data support the interaction between two types of treatments with completely different mechanisms of action enhancing or decreasing each other’s activity. About 83% of patients included in the meta-analysis were on immunotherapy during the interval treatment, potentially creating a favorable setting for TT rechallenge or being influenced by the immune-evasive state promoted after first-line TT. We could not properly assess the influences of targeted therapy on immunotherapy response nor the influences of immunotherapy on TT rechallenge response due to insufficient data, thereby directing this issue for further investigation.

While immunotherapy has been established as the first-line option for treatment-naïve metastatic melanoma patients, including those harboring a BRAF mutation [92,93,94], patients who progress on a targeted therapy and later on immunotherapy, or the inverse order, still have limited treatment options and poor prognosis [88,96,97,98,99,100]. Given this landscape, several case series emerged demonstrating encouraging activity with the re-introduction of targeted therapy (TT) in patients with prior progression on BRAFi/MEKi who went on drug holiday or interval treatment [49,50,51,52]. Therefore, rechallenging patients with targeted therapy could overcome the resistance mechanisms related to the previous progression to a BRAFi alone or the combination of BRAFi/MEKi [49,50,51,52,53,54].

In our analysis, prolonged interval therapy from the first TT exposure until rechallenge (greater than 6 months) was associated with a DCR of almost 70%, which is significantly higher than in patients with a short interval therapy (<6 months, DCR of 56%). This difference may be explained by a reversible mechanism of resistance. Tumors are heterogeneous, and the prevalence of distinct intra-tumoral cellular clones may change as the cell replicates and acquires new mutations, or according to the variations in the tumor microenvironment, or by external pressures, such as systemic or local therapies [75,101,102,103]. Hence, after exposure and initial response, some tumoral clones may develop resistance to TT, allowing them to proliferate and survive, leading to disease progression. Once TT is discontinued and the patient switches to a different interval therapy or drug holiday, external pressure changes, allowing other tumoral clones to arise and proliferate. This process may result in a reversion of BRAFi/MEKi resistance due to the growth of sensible tumor clones, resulting in a de novo response to TT [103].

A network meta-analysis of 15 RCTs evaluating systemic therapy for previously untreated advanced BRAF-mutated melanoma demonstrated that patients treated with BRAF and MEK inhibitors achieved higher ORR compared with BRAF alone (odds ratio of 2.00), and both had superior objective responses compared to other treatments [104]. In line with that, combined TT is the standard of care in current clinical practice due to what is known about resistance to single-agent BRAF inhibition. TT combination is administered to very symptomatic patients with a high burden of disease in the first-line therapy, aiming for a faster response than immunotherapy [11,92]. In contrast, some of the included studies in our meta-analysis did not use combination TT as a first-line treatment, leading to some of the limitations described below.

Of note, in our analysis, about 42% of patients received BRAFi alone and not combination BRAFi/MEKi as the initial TT exposure prior to rechallenge, which may have positively influenced the responses to therapy found in our study. In these patients, it is possible that MEK sensitivity remained intact through a lack of external pressure and those patients could potentially benefit more when treated with the combination of BRAFi/MEKi in the rechallenge setting. Conversely, 21% of patients were also treated with BRAFi alone in the rechallenge setting. Due to lack of patient-level data, we could not explore this hypothesis further or evaluate the magnitude of influence this had on our results. Currently, combined TT is the standard of care for BRAF-mutant melanoma patients [105,106], and the response rates with TT rechallenge in clinical practice may be somewhat lower than what was reported in this paper.

Our population consisted of 53% (164/310) of patients with brain metastasis. We analyzed the presence of brain metastasis as a prognostic factor in addition to the limited data available for this subgroup and did not find a significant association with OS and PFS. The brain is a frequent site of recurrence in BRAF-mutant melanoma patients across different ethnicities [107,108,109]. Studies have indicated a higher incidence of brain metastases in patients treated with BRAF/MEK inhibitors compared to those treated with immunotherapy [107,110]. This highlights the importance of continuous surveillance and management strategies for brain metastases [107,110]. One interesting question is if the recurrence patterns after rechallenge are similar to the first TT exposure. Specifically, understanding brain metastasis response and recurrence would inform us about the central nervous system penetration of TT rechallenge and the need of further local therapy. The information regarding progressive sites of disease after rechallenge was not available in our studies. Thus, we were unable to analyze whether there is a difference in the incidence of CNS disease recurrence and progression between first-line TT therapy and rechallenge.

The LDH level is a well-known factor associated with prognosis in melanoma patients [111,112,113]. In our population, 48.5% (194/400) of patients had high LDH levels. Yet, we could not demonstrate a significant association of LDH with OS and PFS due to a lack of survival data in this subgroup among the studies. Interestingly, regardless of these high-risk features (elevated LDH and brain metastasis), our systematic review and meta-analysis demonstrated benefits in tumor response with TT rechallenge for all population.

None of the studies in our meta-analysis reported any unexpected adverse events when rechallenging advanced melanoma patients with BRAFi/MEKi. This underscores the well-known safety profile of this treatment strategy, which was also observed in the first-line therapy and can be applied to the rechallenge setting [39,114].

## 5. Limitations

Our study has some limitations. We did not pool data on time-to-event survival outcomes, such as PFS and OS, due to the constraints of missing information in the included studies. We managed this by reporting PFS and OS as the median of medians. Even though most of our studies were observational cohort studies and may have selection bias influencing their results, we were able to demonstrate tumor response to TT rechallenge with low heterogeneity in our primary or sensitivity analyses. Notwithstanding, our study presented new and relevant data concerning the benefit of adding a palliative treatment with TT rechallenge for BRAF-mutant metastatic melanoma patients.

## 6. Conclusions

To the best of our knowledge, this is the first systematic review and meta-analysis to pool available data on rechallenging BRAF-mutant advanced melanoma patients with MAPK inhibitors. Our findings indicate that most of patients can benefit from TT rechallenge by achieving the partial control or stability of their disease. Better outcomes were observed in patients with a TT-free interval greater than six months. Additionally, the median overall survival was almost ten months in the third-line setting in all population analyses, including patients with brain metastasis and high LDH.

Our study provided important evidence that rechallenge with BRAFi/MEKi should be considered as a palliative treatment option for patients who progress on MAPKi and had an interval period with or without other treatments. Further research will be important to investigate the patterns of recurrence and progression during rechallenge and the best sequencing strategy to optimize patient response and survival.

## Figures and Tables

**Figure 1 cancers-15-03754-f001:**
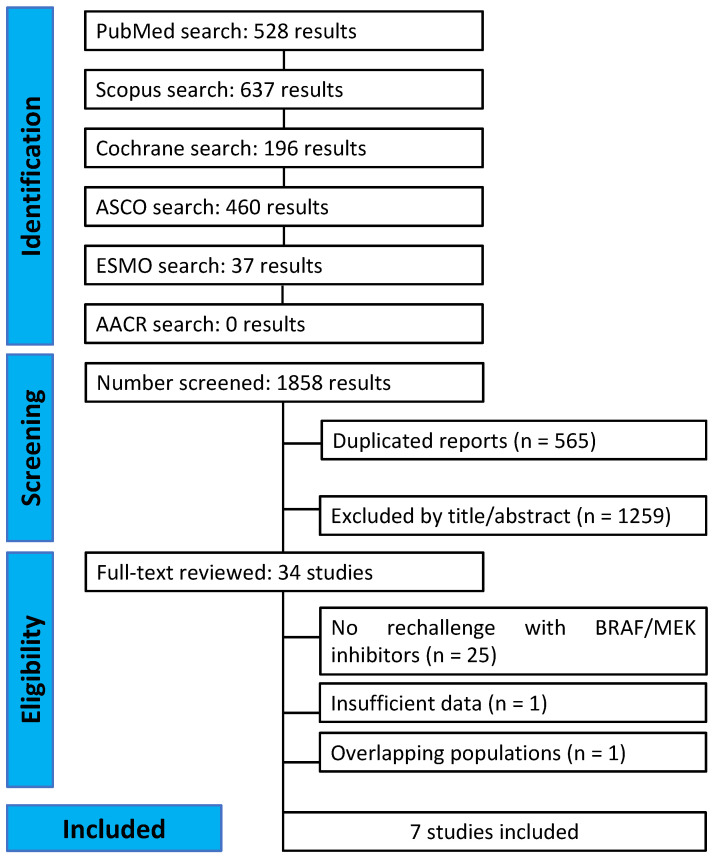
PRISMA flow diagram of the study screening and selection.

**Figure 2 cancers-15-03754-f002:**
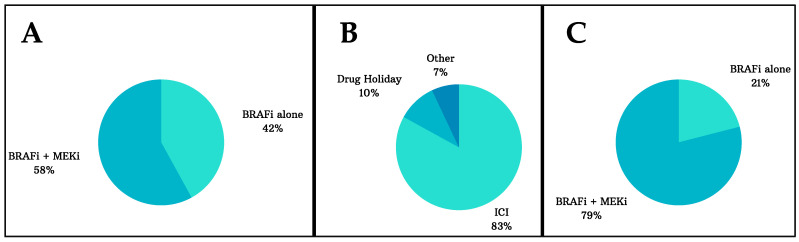
Summary of the treatments administered during the first targeted therapy (TT) exposure (**A**), during the interval period (**B**), and the TT rechallenge period (**C**). During the first TT, most patients (58%) received the combination of BRAFi plus MEKi, and 42% received BRAFi alone. A total of 83% of patients received immunotherapy during the interval treatment, whereas 10% were on a drug holiday. In the rechallenge setting, almost 80% of patients received the combination of BRAFi plus MEKi, while 21% were treated with BRAFi alone.

**Figure 3 cancers-15-03754-f003:**
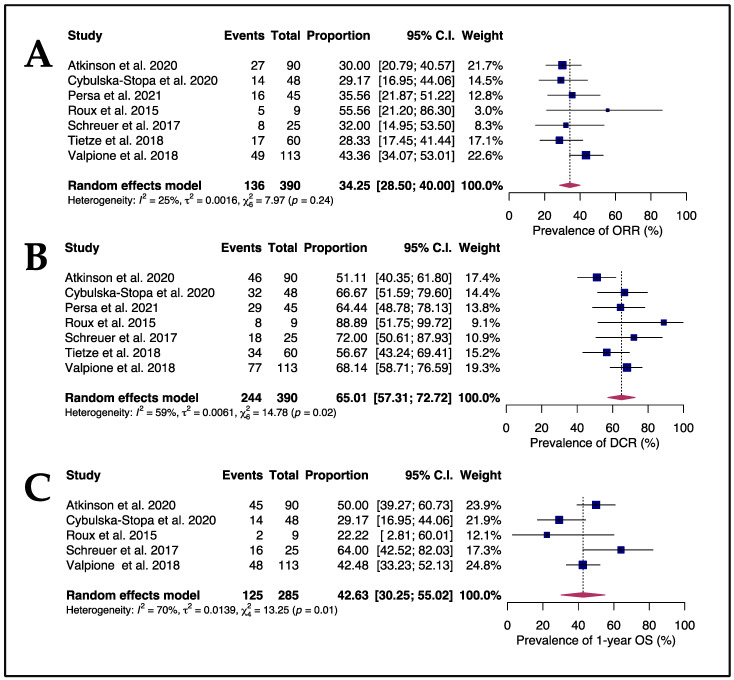
Forest plots showed an ORR of 34.25% (**A**), a DCR of 65.01% (**B**), and a 1-year OS rate of 42.63% (**C**) [60,61,62,63,64,65,66].

**Figure 4 cancers-15-03754-f004:**
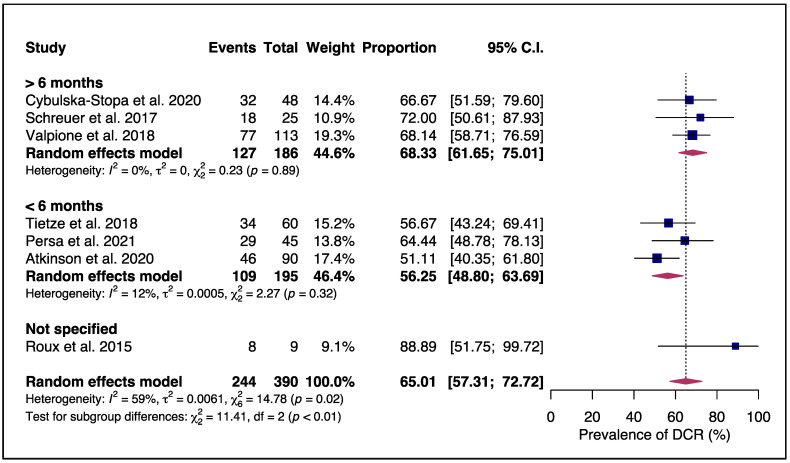
Subgroup analysis of DCR during targeted therapy (TT) rechallenges according to the median BRAFi/MEKi-free interval. Patients with a BRAFi/MEKi-free interval ≥6 months had a DCR of 68.3%, significantly higher than patients with a BRAFi/MEKi-free interval of <6 months, with a DCR of 56.2% (*p* < 0.01) [60,61,62,63,64,65,66].

**Table 1 cancers-15-03754-t001:** Baseline characteristics of the population in the included studies.

Study ID	N	Age ^†^, y	TT in 1st/2nd Line with BRAFi + MEKi (%)	IT with ICI (%)	Interval between TT and RC ^†^ (Months)	RC with BRAFi + MEKi (%)	ECOG (0–1, ≥2) (%)	LDH (Normal, ≥ULN) (%)	CNS Disease at RC (yes, no) (%)
Atkinson (2020) [60]	90	61	80	100	NA	93	60, 28	30, 51	49, 46
Cybulska-Stopa (2020) [61]	51	56	68	100	8.6	96	78, 22	22, 76	59, 41
Persa (2021) [62]	48	57	79	75	4	83	NA	40, 60	50, 50
Roux (2015) [63]	10	52.4	0	80	NA	10	50, 50	70, 30	60, 40
Schreuer (2017) [64]	25	54.7	64	100	6.1	100	80, 20	72, 28	68, 32
Tietze (2018) [65]	60	56	32	67	3.4	68	63, 32	43, 57	60, 40
Valpione (2017) [66]	116	51.9	35	71	7.7	66	61, 22	39, 55	44, 56

N: number. y: year. TT: targeted therapy. IT: interval treatment. ICI: immune checkpoint inhibitors. RC: rechallenge. BRAFi: BRAF inhibitor. MEKi: MEK inhibitor. LDH: lactate dehydrogenase. ULN: upper limit of normality. CNS: central nervous system. NA: not available. ^†^ Median.

## Data Availability

The data for this study were systematically collected and organized into a comprehensive database. Access to the data can be granted upon request from the corresponding author.

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
