# Peer review of "Efficacy and Safety of Rechallenge with BRAF/MEK Inhibitors in Advanced Melanoma Patients: A Systematic Review and Meta-Analysis"

_cancers, 2023, doi:10.3390/cancers15153754_

Round 1
Reviewer 1 Report
This is a well-written systematic review on the topic. The authors are commended for a thorough analysis of published data.
Author Response
"Please see the attachment."

Reviewer 2 Report
The systematic review and meta-analysis aimed to evaluate the efficacy and safety of rechallenging advanced melanoma patients with BRAFi/MEKi should be of interest for the readers of the journal. However, the paper would benefit from revisions. For specific comments see below.
Since this is intended as systematic review the number of references (54) is unsatisfactory. Please at least duplicate this number with relevant references on the topic.
There are different reasons for resistance to therapy, which would require additional comments by the authors. Also I am surprised that melanin pigmentation and melanogenesis as an important variable affecting natural history of the disease (Frontiers in Oncology 2022;12. DOI: 10.3389/fonc.2022.842496) is completely ignored. This should improved.
Additional mechanisms by which tumor can protect itself from therapy or host responses is production of different bioregulatory factors acting on systemic or local levels as recently discussed (Trends Neurosci 46: 263-275, 2023.,https://doi.org/10.1016/j.tins.2023.01.003). Such mechanism would be of interest to the readers and are worthy mentioning.
Finally, section limitations is missing, which would be important as relates to the topic of this manuscript.
Author Response
"Please see the attachment."

Reviewer 3 Report
In this manuscript, authors analyzed the effect of rechallenge with BRAF/MEK inhibitors in advanced melanoma patients using previous studies. Overall, the manuscript is well-written and interesting. The following are my comments.
Authors should analyze the effect of ICI to the effect of rechallenge with BRAF/MEK inhibitors. In effective cases of ICI, did the rechallenge with BRAF/MEK inhibitors also show more anti-tumor effect?
Brain is the frequent recurrence site after BRAF/ MEK inhibitors as reported in PMID: 33503528, PMID: 32602174. Authors should analyze if there is difference of recurrent site between initial BRAF/ MEK inhibitors and rechallenge with adding these references.
Author Response
"Please see the attachment."

Round 2
Reviewer 2 Report
The authors adequately revised the manuscript
Reviewer 3 Report
The manuscript was significantly modified, and I have no more comments.